# *coiaf*: Directly estimating complexity of infection with allele frequencies

**Aris Paschalidis**[1][�], **Oliver J. Watson**[1,2][�], **Ozkan Aydemir**[1,3], **Robert Verity**[2], **Jeffrey A. Bailey**[1]*

**1** Department of Pathology and Laboratory Medicine, Brown University, Providence, Rhode Island, United States of America, **2** Department of Infectious Disease Epidemiology, Imperial College London, London, United Kingdom, **3** Program in Molecular Medicine, University of Massachusetts Chan Medical School, Worcester, Massachusetts, United States of America

☉ These authors contributed equally to this work.
* jeffrey_bailey@brown.edu

**Data Availability Statement:** Documentation for coiaf can be found at https://bailey-lab.github.io/coiaf. Additional code used to analyze data and create figures can be found at https://github.com/bailey-lab/coiaf-manuscript-work (DOI: 10.5281/

## Abstract

In malaria, individuals are often infected with different parasite strains. The complexity of infection (COI) is defined as the number of genetically distinct parasite strains in an individual. Changes in the mean COI in a population have been shown to be informative of changes in transmission intensity with a number of probabilistic likelihood and Bayesian models now developed to estimate the COI. However, rapid, direct measures based on heterozygosity or *FwS* do not properly represent the COI. In this work, we present two new methods that use easily calculated measures to directly estimate the COI from allele frequency data. Using a simulation framework, we show that our methods are computationally efficient and comparably accurate to current approaches in the literature. Through a sensitivity analysis, we characterize how the distribution of parasite densities, the assumed sequencing depth, and the number of sampled loci impact the bias and accuracy of our two methods. Using our developed methods, we further estimate the COI globally from *Plasmodium falciparum* sequencing data and compare the results against the literature. We show significant differences in the estimated COI globally between continents and a weak relationship between malaria prevalence and COI.

## Author summary

Computational models, used in conjunction with rapidly advancing sequencing technologies, are increasingly being used to help inform surveillance efforts and understand the epidemiological dynamics of malaria. One such important metric, the complexity of infection (COI), indirectly quantifies the level of transmission. Existing "gold-standard" COI measures rely on complex probabilistic likelihood and Bayesian models. As an alternative, we have developed the statistics and software package *coiaf*, which features two rapid, direct measures to estimate the number of genetically distinct parasite strains in an individual (the COI). Our methods were evaluated using simulated data and subsequently compared to current state-of-the-art methods, yielding comparable results. Lastly, we

zenodo.7931661) and https://github.com/bailey-lab/coiaf-real-data (DOI: 10.5281/zenodo.7826507).

**Funding:** All authors (AP, OJW, OA, RV, JAB) acknowledge funding from the National Institutes of Health (NIH) and the National Institute of Allergy and Infectious Diseases (NIAID) (reference R01AI139520). RV additionally acknowledges funding from the MRC Centre for Global Infectious Disease Analysis (reference MR/R015600/1), jointly funded by the UK Medical Research Council (MRC) and the UK Foreign, Commonwealth & Development Office (FCDO), under the MRC/FCDO Concordat agreement and is also part of the EDCTP2 programme supported by the European Union. OJW was further supported by a Schmidt Science Fellowship in partnership with the Rhodes Trust. The funders had no role in study design, data collection and analysis, decision to publish, or preparation of the manuscript.

**Competing interests:** The authors have declared that no competing interests exist.

examined the distribution of the COI in several locations across the world, identifying significant differences in the COI between continents. *coiaf*, therefore, provides a new, promising framework for rapidly characterizing polyclonal infections.

This is a *PLOS Computational Biology* Methods paper.

## Introduction

Malaria remains a leading cause of death worldwide—in 2021, there were an estimated 247 million cases and 619,000 deaths around the globe [1]. Despite the considerable burden of malaria, these numbers represent the substantial global progress made to control malaria in the last two decades. The WHO reports that 2 billion malaria cases and 11.7 million malaria deaths were averted globally from 2000 to 2021 [1]. The majority of these gains reflect an increase in vector control initiatives [2–4], the development of highly efficacious antimalarial combination therapies [5–7], and improved case management through the deployment of rapid diagnostic tests (RDTs) [8–13]. However, evidence indicates that progress has slowed and that there is a need for new approaches to capitalize on the gains already made [1].

One approach is the use of computational methods, which often rely on recent advances in genetic sequencing and provide an increased understanding of malaria biology, to help inform control efforts [14–16]. For example, molecular and genomic epidemiology surveillance tools, which have been developing rapidly, can track drug resistance and understand and evaluate control efforts [17, 18]. Moreover, computational methods have been applied to identify polyclonal infections—malaria infections with multiple distinct strains [19, 20]. Such polyclonal infections introduce additional genetic complexity that is often difficult to account for computationally. As a result, many of the population genetic tools frequently applied to study other organisms are unsuitable for studying malaria, and researchers often rely on limiting genetic analyses to individuals who are monoclonally infected [21].

Although determining the most informative metrics is an active field of investigation [22], one important metric is the complexity of infection (COI). Sometimes referred to as multiplicity of infection, although this is generally reserved for infections within the same cell, the COI represents the number of genetically distinct malaria genomes or strains that can be identified in a particular individual. These polyclonal infections may arise from one or both of the following: (*i*) a single infectious mosquito feeds on a human host, transferring several genetically distinct parasite strains (often referred to as a co-transmission event [23, 24]) or (*ii*) two or more infectious mosquitoes with distinct malaria strains feed on an individual (known as a superinfection event [24, 25]). Measures of genetic diversity and the COI are increasingly used for inferring malaria transmission intensity and evaluating malaria control interventions [18]. Transmission intensity has been shown to impact the contribution of each event towards the generation of within-host parasite genetic diversity [22]. Superinfection is modulated by the host's current infections [26], age, and exposure-acquired immunity [27]. Additionally, the COI provides a practical approach for identifying monoclonal infections to simplify genomic analyses.

Traditionally, the COI was measured in one or a few regions of the genome, relying on the enumeration of the maximal number of haplotypes detected through PCR amplification at

genes or markers encoding highly diverse length polymorphisms. Two of the most common markers are the merozoite surface proteins 1 and 2 (*msp1* and *msp2*), surface proteins found on the merozoite stage of the malarial parasite [28, 29]. Traditional methods are hindered by limitations on the number of loci examined [30, 31], and lack the ability to detect parasites at low parasitemia in samples [32]. Furthermore, sanger sequencing lacks sensitivity for low parasitemia samples without employing laborious subcloning [33].

High-throughput sequencing provides more sensitive and specific methods. As a result, new computational methods have been developed to determine the COI. Two early proposed methods were the *FwS* metric by Aubern et al., which characterizes within-host diversity and its relationship to population-level diversity [30], and the *estMOI* software, which utilizes local phasing information of microhaplotypes within read pairs to estimate the COI [32]. Unfortunately, the *FwS* metric does not directly relate to the COI nor has a concrete biological interpretation, and the *estMOI* software relies on observed local haplotypes and heuristic interpretation. More recently, new tools have been developed to better measure the COI beyond the maximal observed haplotype. For instance, the *DEploid* software package uses haplotype structure to infer the number of strains, their relative proportions, and the haplotypes present in a sample [34]. However, it is known that *DEploid* under-predicts the COI for high COI infections [24]. Other methods have been developed to examine the relatedness between parasite strains [35, 36]. The current state-of-the-art method for determining the COI of a sample is *THE REAL McCOIL*, which is an extension of the *COIL* method [31]. *THE REAL McCOIL* employs a Bayesian approach, turning heterozygous data into estimates of allele frequency using Markov chain Monte Carlo methods and jointly estimating the likelihood of the COI [37].

Despite various methods for estimating the COI, no rapid, direct measures have been developed to work effectively on a set of loci or at the genome-wide level. In this work, we present two new methods that use easily calculable metrics to directly estimate the COI from allele frequency data. Our two methods closely resemble the categorical and proportional methods implemented in *THE REAL McCOIL* [37], yet are geared towards large numbers of loci and provide rapid estimates of the COI. Using our methods, we have developed the software package *coiaf* in the programming language ℝ, a language and environment for statistical computing and graphics [38].

## Materials and methods

### Problem formulation

Current state-of-the-art approaches for estimating the COI rely on identifying the number of different parasites present in an infection using high-throughput sequencing. In monoclonal infections, i.e., infections composed of only one parasite strain (COI = 1), all sequence reads should be identical, originating from the same parasite strain. However, a combination of each parasite strain, proportional to the strain's abundance, will contribute to the observed sequence reads in mixed infections. At genetic loci containing variation, there is an increased chance of observing multiple alleles as the number of unrelated parasite strains within an infection increases. Therefore, the likelihood of observing multiple alleles at any locus depends on the number of parasite strains in an infection and the prevalence of genetic polymorphisms in the population.

We focus on only biallelic SNPs—the vast majority of loci—and define the major allele as the most prevalent allele in a population. We note that any multiallelic site can be collapsed into a biallelic site, although some information will be lost. Assuming for any individual there are *l* biallelic loci, we define the population-level allele frequency (PLAF) as the mean within-sample allele frequencies (WSAF) of the alternate allele, i.e., the non-3D7 reference, at each

locus across a population. We further introduce the population-level minor allele frequency (PLMAF), which first requires us to define the minor allele at each locus. The minor allele is defined as the alternate allele if PLAF $\leq 0.5$, and the reference allele if PLAF $> 0.5$. The PLMAF is defined as a vector, **p**, of length $l$ composed of the frequencies of the minor allele at each locus across a population, namely **p** = $(p_1, \ldots, p_l)$, where $p_i \in [0, 0.5]$. Additionally, we define the within-sample minor allele frequency (WSMAF) as a vector, **w**, of length $l$ composed of the frequencies of the population-level minor allele at each locus for a single individual infection. For instance, the WSMAF will be equal to one when all sequence reads observed at a given locus are of the population-level minor allele.

## Variant and Frequency Methods

**Variant Method.**   Our overall goal is to estimate the COI of the sample, denoted by $k$, using the WSMAF and the PLMAF. We do this by comparing observed data to derived expressions that define a relationship between the WSMAF, the PLMAF, and the COI. We present two alternative expressions that we refer to as the Variant Method and the Frequency Method.

In the Variant Method, we examine a set of SNPs and express the probability of SNP $i$ being heterozygous with respect to the PLMAF and the COI. We define $V_i$, a Bernoulli random variable that takes the value of one if a site is heterozygous and zero otherwise. The probability that locus $i$ is heterozygous, written as $\mathbb{P}(V_i = 1)$, will be equal to one minus the probability that a locus is homozygous (see Appendix B in S1 Appendices). We thus write,

$$\mathbb{E}[V_i] = \mathbb{P}(V_i = 1) = 1 - p_i^k - (1 - p_i)^k. \tag{1}$$

As the COI increases, the probability of observing a heterozygous locus within an infection also increases (see Fig 1). We note that this is the same expression used within the categorical method of *THE REAL McCOIL* (Eq (2)) [37]. Similarly to the categorical method of *THE REAL McCOIL*, we assume that loci are independent. If this assumption is not met, the confidence intervals around COI estimates will be impacted with bootstrapped confidence intervals increasing, but the bias in estimated COI should not change, providing the model is otherwise well specified (see Appendix C.2 in S1 Appendices). This is the same outcome as for the *THE REAL McCOIL* categorical method [37], and first described in the earlier *COIL* method [31].

**Frequency Method.**   In the Frequency Method, we focus on the expected value of the within-sample minor allele frequency. For the sake of simplicity, the complete derivation has been left to Appendix B in S1 Appendices. Briefly, we determine the probability of a particular strain carrying the minor allele and then determine the expected WSMAF by summing over the expected contribution of each strain. We represent the expected value of the within-sample minor allele frequency given that a site is heterozygous as follows:

$$\mathbb{E}[W_i | V_i = 1] \quad = \frac{p_i - p_i^k}{1 - p_i^k - (1 - p_i)^k} . \tag{2}$$

## Estimation method

Given data, $D : \{(p_i, w_i, d_i), i = 1, \ldots, l\}$, where $p_i$ is the PLMAF at locus $i$, $w_i$ is the WSMAF at locus $i$, and $d_i$ is the within-sample sequencing coverage at locus $i$, we next explore our methods to approximate the COI of a sample. Data are first processed to account for sequencing error. This process denotes loci at which there was suspected sequencing error as homozygous instead of heterozygous (for additional information, see Appendix D in S1 Appendices).

Following adjustment for sequence error, we consider an arbitrary data point $(p_i, w_i, d_i)$. Recall that the Variant Method and the Frequency Method examine different random

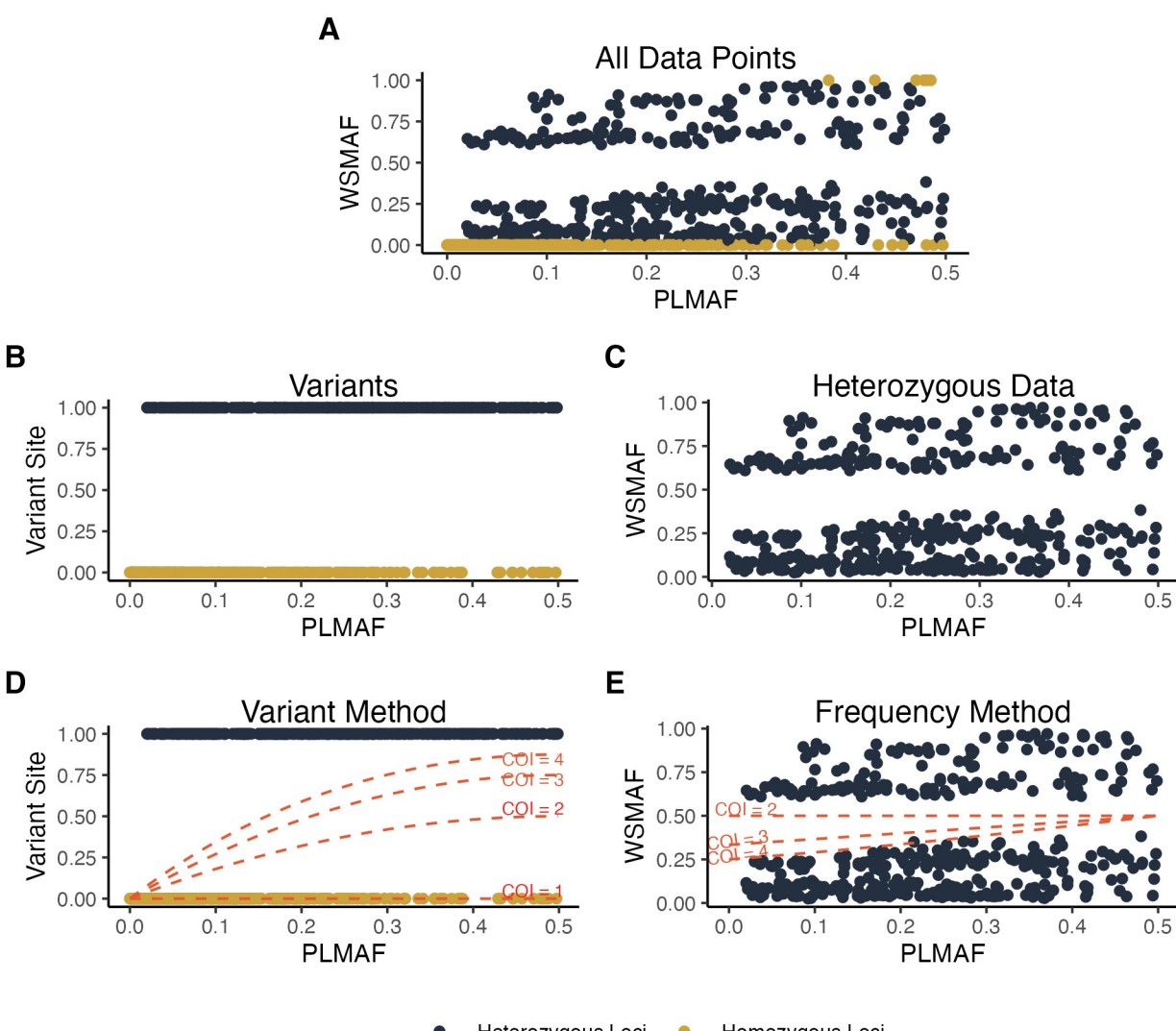

**Fig 1. Flowchart of methods. (A)** The relationship between the WSMAF and the PLMAF is shown for an example simulation with a COI of 4.
**(B)** Data have been processed so that loci are deemed variant if they are heterozygous and invariant otherwise. **(C)** Homozygous data have been
filtered out. **(D-E)** Following the processing of data, Eqs (1) and (2) have been plotted for varying COIs from 1 to 4, respectively.

variables. Specifically, the Variant Method identifies the probability of a locus being hetero-
zygous, $\mathbb{P}(V_i = 1)$, and the Frequency Method identifies the expected value of the WSMAF
given a site is heterozygous, $\mathbb{E}[W_i | V_i = 1]$. To determine the COI, we utilize Eqs (1) and (2).
We solve the following weighted least squares minimization problem for the Variant
Method:

$$\min_{k} \left( \sum_{i=1}^{l} (v_i - (1 - p_i^k - (1 - p_i)^k))^2 d_i \right), \tag{3}$$

where $v_i$ is defined as the value of the random variable $V_i$. Similarly, we solve the following

weighted least squares minimization problem for the Frequency Method:

$$\min_k \left( \sum_{i=1}^{l} \left( v_i \left( w_i - \left( \frac{p_i - p_i^k}{1 - p_i^k - (1 - p_i)^k} \right) \right) \right)^2 d_i \right). \tag{4}$$

Note that the estimation methods described minimize the sum of squared residuals between the observed data and the relationships derived in Eqs (1) and (2).

**Solution methods.** We solve this optimization problem using two methods: (*i*) assuming discrete values of the COI and (*ii*) assuming continuous values of the COI. Recall that COI is defined as the number of genetically distinct malaria parasite strains an individual is infected with. While a continuous value of the COI has no direct biological interpretation, significant departures from discrete values are expected due to parasite relatedness and a range of other biological phenomena, including, but not limited to, overdispersion in sequencing and parasite densities. Therefore, a continuous value of the COI may provide a more accurate representation of the overall population of samples being studied. Furthermore, as relatedness in mixed infections is common [36], a continuous COI may provide insights into the degree of relatedness between parasite strains in mixed infections and detect highly-related polyclonal infections that may traditionally be categorized as monoclonal.

We use a brute force approach to solve the discrete versions of the previously defined optimization problems, which involves computing the objective function for each COI considered. As brute force approaches can be computationally inefficient, we limit the range of values of the COI. To solve the continuous versions of the optimization problems, we utilize ℝ's built-in optimization function [38]. In particular, we leverage a quasi-Newton L-BFGS-B approach with box constraints [39]. We set the lower and upper bounds of the COI as 1 and 25, respectively, with the default starting value of the COI equal to two. Note that in both the discrete and continuous case, the upper bound of the COI is much larger than most COIs seen in the real world [37, 40]. In both cases, we also provide the capability to determine the 95% confidence interval for our COI estimates by leveraging bootstrapping techniques [41] (see Appendix E in S1 Appendices for more details).

## Data

To evaluate the accuracy and sensitivity of our methods, we created a simulator that generates synthetic sequencing data for several individuals in a given population. In overview, each individual is assigned a COI value. The haplotype of each strain is then assigned by sampling from the population-level minor allele frequency. Next, we simulate the number of sequence reads mapped to the reference and alternative allele by sampling in proportion to the parasite densities for each strain. After simulating sequence error, the mapped sequence reads are then used to derive the within-sample minor allele frequency. A detailed description of our simulator can be found in Appendix G in S1 Appendices.

In addition to simulated data, we use sequencing data sampled from infected individuals worldwide to compare our methods to the current state-of-the-art COI estimation metric and investigate the COI distribution across the world. We analyzed over 7,000 *P. falciparum* samples from 28 malaria-endemic countries in Africa, Asia, South America, and Oceania from 2002 to 2015 from the MalariaGEN *Plasmodium falciparum* Community Project [42]. Detailed information about the data release, including brief descriptions of contributing partner studies and study locations, is available in the supplementary of MalariaGEN et al. [42]. We used the provided variant call files (VCFs) generated using a standard analysis pipeline. The median read depth of coverage of the initially sequenced field isolates was 73 across all samples. After

removing replicate samples, mixed-species samples, and samples with low coverage, suspected contamination, or mislabelling, 5,970 samples remained for further analysis. Genomic data were further filtered for high-quality biallelic coding and non-coding SNPs as outlined in Zhu et al. [36]. Additionally, data were filtered to sites that are part of the core genome.

To apply our developed methods, we must estimate the population-level frequency of the minor allele. Consequently, we sought to assign the samples to a suitable number of geographic regions such that the number of samples per region was suitable for the reliable estimation of the population-level minor allele frequencies. We used the Partitioning Around Medoids (PAM) algorithm to solve a k-medoids clustering problem [43, 44] to group samples based on the longitude and latitude of sample collection. We next calculated the silhouette information for each clustering of k groups [45], arriving at 24 regions globally (see Appendix L.2 in S1 Appendices for a map of locations). Given these 24 clusters, we filtered SNPs to variants with a population-level alternative allele frequency greater than 0.005 in each region. The 0.005 frequency cutoff was chosen as sequence error likely obscures the detection of true variation from parasite strains comprising less than 0.5% of total parasite density. Clusters of data were additionally traced to a specific continent and subregion as defined by the World Development Indicators [46, 47].

## Results

### Performance on simulated data

We simulated data for 1,000 loci with a read depth of 200 at each locus using our simulator. Data were simulated with a complexity of infection ranging from 1 to 20. This simulation did not introduce error to determine optimal performance based on sampling. Our methods, therefore, accounted for no sequencing error. The results of running the discrete version of the Variant Method and the Frequency Method are illustrated in Fig 2A and 2B, respectively.

The Variant Method and the Frequency Method perform well for all COIs between 1 and 20. Notably, the lower the true value of the COI, i.e., the COI that data was simulated with, the better our models perform with a mean absolute error close to 0. Our models exhibit more variability across subsequent iterations as the COI increases and underestimate the true COI. For example, at a COI of 20, the estimated COI from the Frequency Method ranges from 13 to 24 —the maximum COI our model could output was 25 in these trials. Nevertheless, most predicted COIs remain close to the true COI as witnessed by the low mean absolute errors of at most 2.14 in Fig 2C. Comparing our two methods, we find that the Variant and the Frequency Methods perform equally with insignificantly different mean absolute errors (p-value = 0.174) and biases (p-value = 0.884).

### Sensitivity analysis

To understand the sensitivity of our models to alterations in the parameters considered, we tested the performance of the discrete and continuous representations of the Variant Method and the Frequency Method, assessing changes in the accuracy of our predictions. For each sample, we utilized bootstrapping techniques [48, 49] to determine the mean absolute error and bias of the predicted COI compared to the true COI. Furthermore, we ran each algorithm several times to ensure reliable results. A description of several key parameters perturbed and their default values can be found in Appendix H in S1 Appendices. The resulting figures can be found in Appendix L.1 in S1 Appendices.

Here, we highlight the effect of varying two metrics that can be controlled in the field: the read depth at each locus and the number of loci sequenced. Sequencing more loci at larger read depths is preferred as this results in higher-quality data. In general, as the coverage at

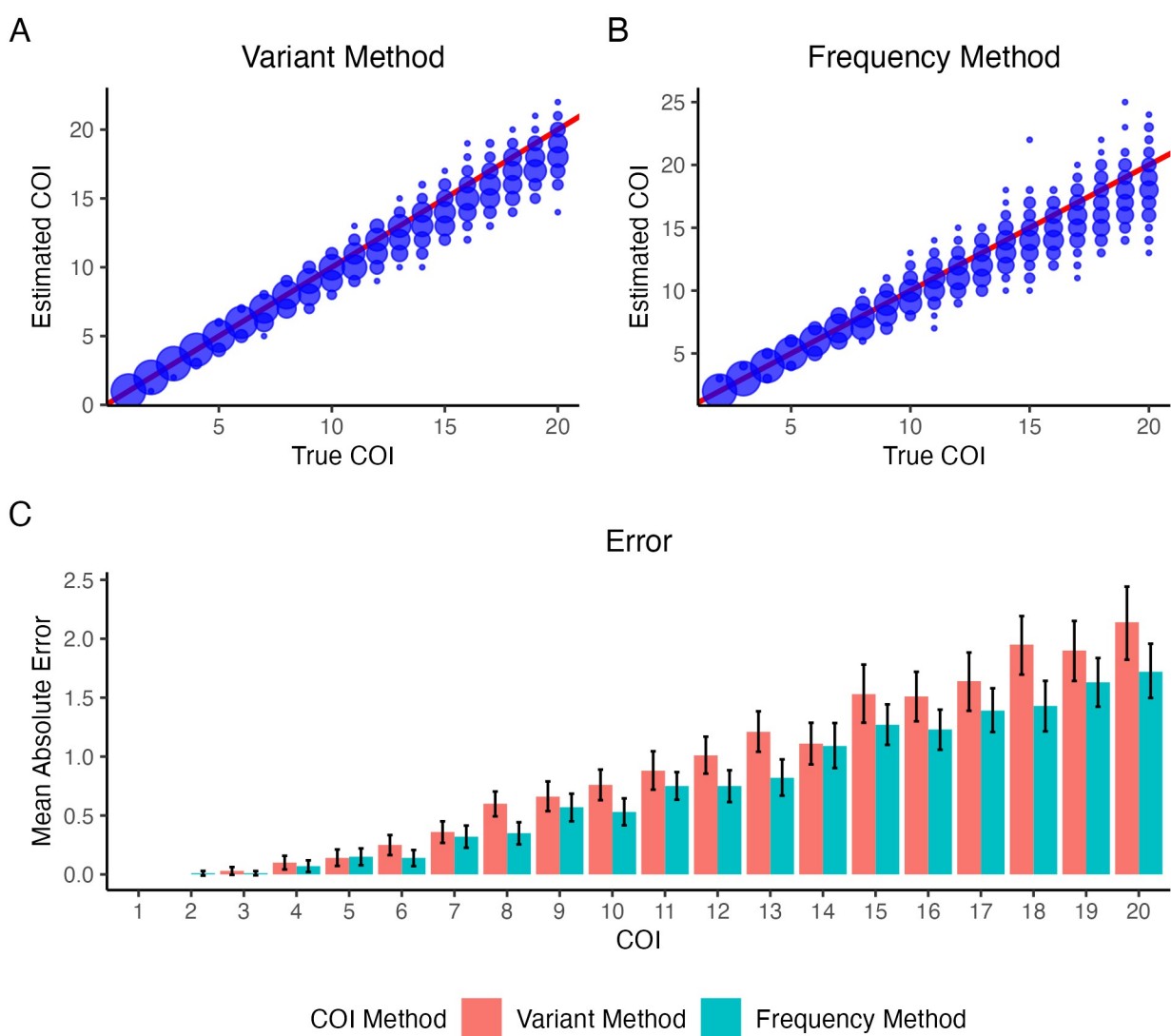

**Fig 2. Estimating the COI on simulated data.** The performance of the Variant Method (**A**) and Frequency Method (**B**) is shown for 100 simulations of a COI of 1–20 with 1,000 loci, a read depth of 200, no error added to the simulations, and no sequencing error assumed. Point size indicates density, with the red line representing the line $y = x$. (**C**) The mean absolute error for each method is shown. The black bars indicate the 95% confidence interval.

each locus increases, the performance of our methods also improves (see Fig K and L in S1 Appendices). A monotonic, non-linear relationship is observed between sequence coverage and the mean absolute error, with diminishing returns in performance observed with sequence coverage greater than 100. As was the case for our coverage data, when the number of loci sequenced is low, around 100 loci, our methods have high variability and tend to underpredict the true COI (see Fig M and N in S1 Appendices). However, the performance increases as the number of loci increases to 1,000. In addition, increasing the number of loci examined above a certain threshold, in this case, 1,000 loci, does not seem to substantially impact the performance of our models. However, note that an increase in the number of loci does reduce our estimate's variance. This reduction in variance will continue to decrease as the number of loci included increases. However, this is dependent on whether loci are independent of one another (see Appendix C.2 in S1 Appendices).

The performance of our methods was also impacted by parameters controlling the underlying malarial biology. In particular, as the relatedness between parasite strains increased, our estimation methods began to underpredict the true COI. However, with 10% relatedness between strains in mixed infections, our methods could still estimate the COI of up to 10 with a low mean absolute error of less than 1. Under prediction of the COI increased consistently with increasing relatedness, with a 50% relatedness resulting in the COI being similarly underestimated by 50% (see Fig R in S1 Appendices). Our methods also underpredicted the COI in simulations with overdispersed parasite densities (see Fig O and P in S1 Appendices), i.e., how uneven parasite densities are in mixed infections, and in simulations with overdispersed read counts (see Fig Q in S1 Appendices), i.e., the observed within-sample minor allele frequency exhibited greater variance than simply being described by a Binomial distribution determined by the true within-sample minor allele frequency.

## Comparison to state-of-the-art methods

In this section, we compare our novel methods to the current state-of-the-art method used to estimate the COI, *THE REAL McCOIL* [37]. To compare these methods, we simulated data across a range of COIs multiple times and evaluated how perturbing parameters influenced the accuracy of both *THE REAL McCOIL* and our methods. We performed five separate analyses, varying the coverage, the number of loci, the level of overdispersion in parasite density, the relatedness between parasite strains, and the error introduced into the simulated data. The results are presented in Appendix L.1.1 in S1 Appendices. Across each analysis, we found few differences between our software and *THE REAL McCOIL*. As the number of loci and the coverage increased, the performance of both software packages improved. Conversely, as greater levels of overdispersion, relatedness, and sequence error were introduced into the simulations, both methods underpredicted the true COI.

Notably, *coiaf* provided improvements in computational performance compared to *THE REAL McCOIL* (see Appendix I in S1 Appendices). When the run time of the two estimation methods was directly compared on simulated data, the speed of the point estimate generated by the discrete and continuous methods of *coiaf* remained constant even as the number of loci increased. Conversely, the speed of *THE REAL McCOIL* increased linearly. As previously mentioned, our software package provides the capability to estimate the 95% confidence interval for our COI estimates using a bootstrapping approach. When comparing *THE REAL McCOIL* against our methods with 100 bootstrap replicates, which was observed to be adequately large to sufficiently capture the uncertainty in the COI (see Appendix F in S1 Appendices), both methods exhibited linear increases in computational time with increasing samples and loci. However, the linear increase in computational time associated with our methods was less than exhibited by *THE REAL McCOIL*.

We additionally compared our methods to the current state-of-the-art method by examining estimates of the COI on real-world data. As previously described, we grouped our data into 24 regions worldwide. To estimate the COI for each of the 5,970 samples, we examined an average of 32,362 (range: 15,276—40,272) loci in each region (see Appendix J in S1 Appendices). Furthermore, we ran 5 repetitions of the *THE REAL McCOIL* on each sample, with a burn-in period of 1,000 iterations followed by 5,000 sampling iterations, and using standard methodology to confirm convergence between Monte Carlo Markov chains [50]. For additional information, see Appendix K in S1 Appendices. Fig 3 examines the COI estimation of *THE REAL McCOIL* and *coiaf* for all samples. We note that the relationship between *coiaf*'s estimated COI and *THE REAL McCOIL*'s estimated COI for each of the 24 individual regions is shown in Fig AF and AG in S1 Appendices.

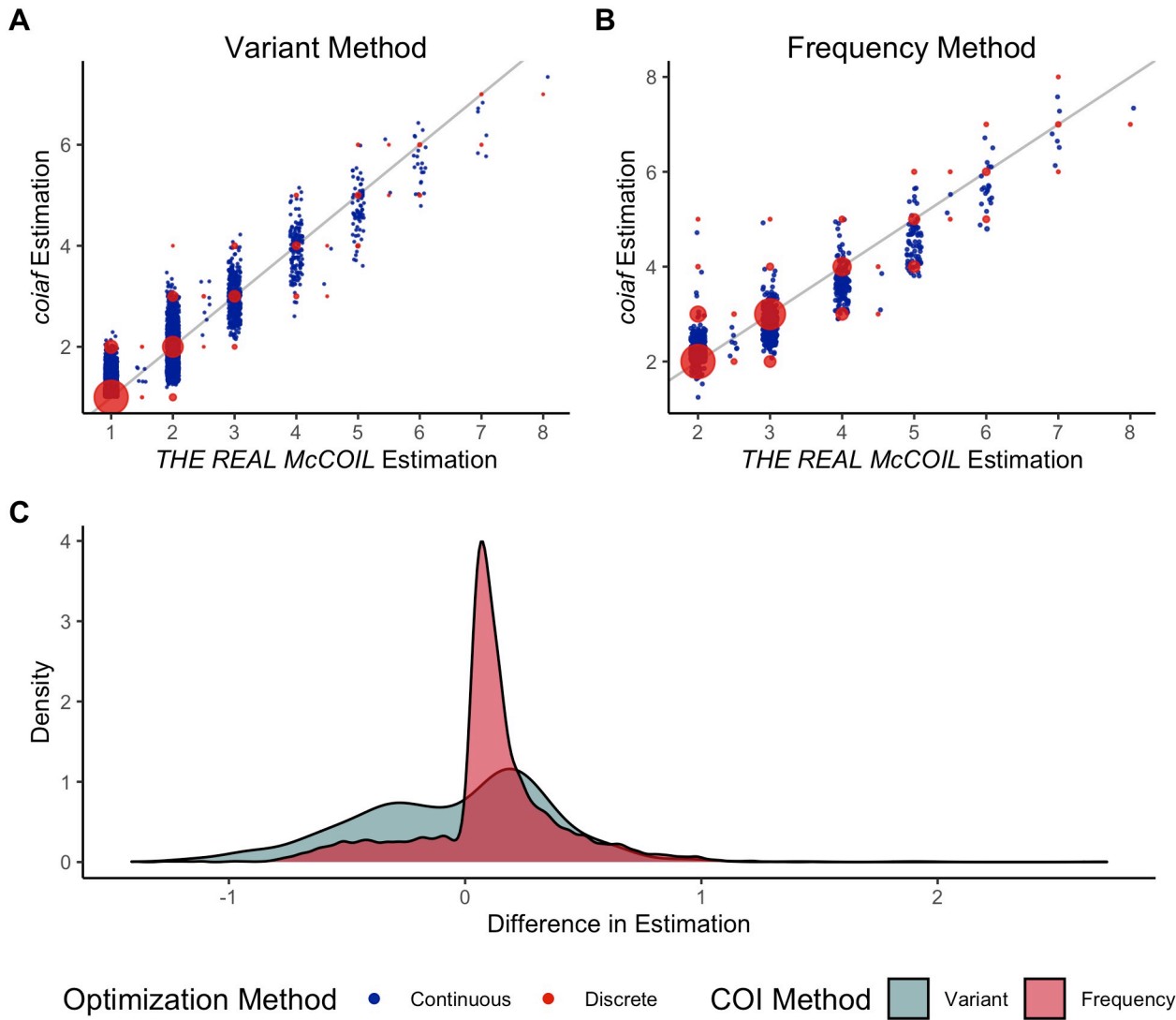

**Fig 3. Comparison between *THE REAL McCOIL* and *coiaf*.** The COI estimation using (**A**) the Variant Method and (**B**) the Frequency Method is compared against the *THE REAL McCOIL*. (**C**) The distribution of differences between our estimation and *THE REAL McCOIL*'s estimation is shown. This difference is computed by subtracting the *THE REAL McCOIL*'s median estimation of the COI from our estimated value of the COI. The high density observed above 0 for the Frequency Method occurs because the Frequency Method is undefined for a COI of 1. Consequently, for samples that *THE REAL McCOIL* estimates as having a COI equal to 2, the distribution of our estimates of the COI using the Frequency Method is skewed greater than 2 (**B**), in contrast to the Variant Method, which exhibits lower skewness (**A**).

We observe that the Variant Method and the Frequency Method strongly correlate with the estimates from *THE REAL McCOIL* (Fig 3A and 3B). When the COI is estimated to be below 5, both methods estimate COI values close to one another. However, as the estimated COI increases, there is greater variability in predictions. At these high COI values, our methods tend to estimate the COI within 3 of *THE REAL McCOIL*'s estimate (Fig 3C). As expected, the continuous estimation methods align with the discrete estimation methods. Furthermore, we note that the Frequency Method does not show estimates when *THE REAL McCOIL* predicts a COI of 1. This is because the Frequency Method at a COI of 1 is undefined; at a COI of 1, there would be no heterozygous loci used in the Frequency Method. We fit linear regression models to the data to quantify the relationship between our novel software and the current state-of-

**Table 1. Relationship between *coiaf* and *THE REAL McCOIL*.** A linear regression model was fit to the data to evaluate the relationship between *coiaf*'s and *THE REAL McCOIL*'s estimation methods. Furthermore, the Pearson correlation between the estimated COIs was computed.

| *coiaf* Estimation Method | Linear Regression | | Correlation |
|---|---|---|---|
| | $R^2$ | P-value | |
| Discrete Variant Method | 0.840 | <0.001 | 0.916 |
| Continuous Variant Method | 0.883 | <0.001 | 0.940 |
| Discrete Frequency Method | 0.804 | <0.001 | 0.896 |
| Continuous Frequency Method | 0.844 | <0.001 | 0.919 |

the-art method and evaluated the Pearson correlation between estimation methods. The results are reported in Table 1 and indicate that each of the methods introduced in *coiaf* is highly correlated with *THE REAL McCOIL*.

## Mapping COI worldwide

To demonstrate *coiaf*'s utility and better understand global patterns of the COI, we examined the distribution of COI in 24 different regions. In each region, we studied an average of 248 (range: 29 to 909) samples and 32,362 loci (range: 15,276 to 40,272) (see Appendix J in S1 Appendices). Our samples can be traced to four different continents, with most originating in either Africa (55.5%) or Asia (41.8%). All other samples were sequenced in Oceania (2.03%) or the Americas (0.620%).

Fig 4 highlights the mean and median COI across all samples in each of the 24 regions outlined previously. We, furthermore, aimed to understand the relationship between the complexity of infection and malaria prevalence by leveraging estimates of malaria microscopy prevalence in children aged two to ten generated by the Malaria Atlas Project [8, 9, 51]. Fig 4C plots the density of the COI for each region sorted by the region's malaria prevalence. Table 2 outlines the mean COI in each of the four continents and seven subregions analyzed.

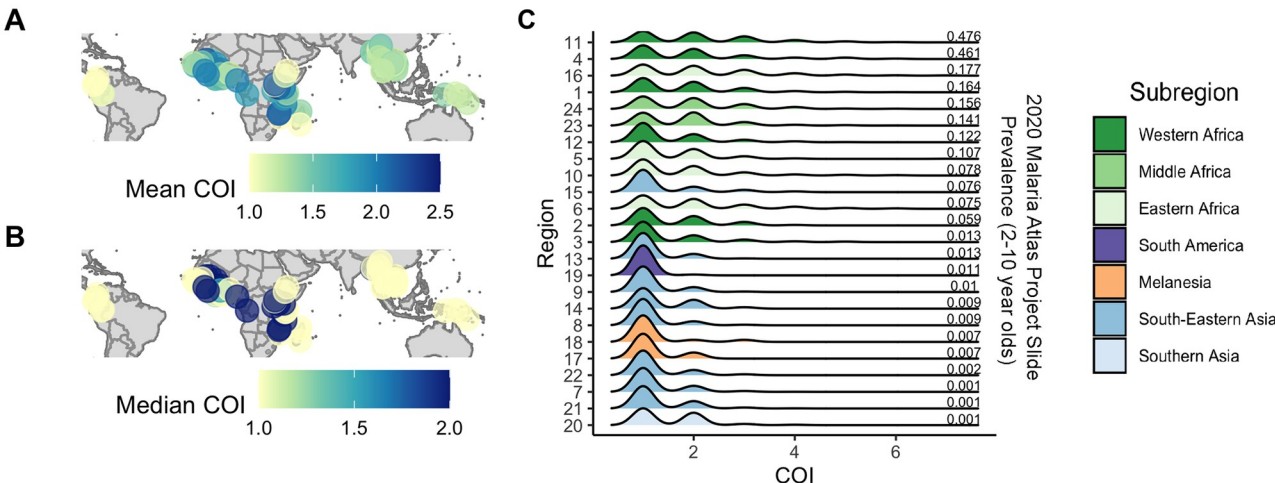

**Fig 4. COI across the globe.** The mean (**A**) and median (**B**) COI of all samples in each study location within the 24 regions is plotted. The color and size of each point represent the magnitude of the COI. (**C**) A density plot for each region, where the color of the plot indicates in what subregion the data was sampled. The plots are sorted by the median microscopy prevalence in children aged two to ten as estimated in the Malaria Atlas Project [8, 9, 51] and indicated to the right of each density plot. Map data was obtained from Natural Earth (medium scale data, 1:50m), which is in the public domain.

**Table 2. Mean COI.** Mean COI across each continent and subregion analyzed.

| Continent | Subregion | Number of Samples | Mean COI (SD) |
|---|---|---|---|
| Africa | Eastern Africa | 739 | 1.88 (1.10) |
| Africa | Middle Africa | 579 | 1.87 (0.948) |
| Africa | Western Africa | 1,996 | 1.87 (1.05) |
| Americas | South America | 37 | 1.03 (0.164) |
| Asia | South-Eastern Asia | 2,421 | 1.25 (0.506) |
| Asia | Southern Asia | 77 | 1.52 (0.641) |
| Oceania | Melanesia | 121 | 1.20 (0.440) |

**Table 3. Relationship between *coiaf* and malaria prevalence.** A linear regression model was fit to the data to evaluate the relationship between COI and prevalence. Furthermore, the Pearson correlation was examined.

| *coiaf* Estimation Method | Linear Regression | | Correlation |
|---|---|---|---|
| | $R^2$ | P-value | |
| Discrete Variant Method | 0.0770 | <0.001 | 0.278 |
| Continuous Variant Method | 0.0787 | <0.001 | 0.281 |
| Discrete Frequency Method | 0.0190 | <0.001 | 0.138 |
| Continuous Frequency Method | 0.0218 | <0.001 | 0.148 |

Of all the continents, Africa had the highest mean COI of 1.87, followed by Asia with a mean COI of 1.26, Oceania with a mean COI of 1.20, and the Americas with a mean COI of 1.03. A Nemenyi post-hoc test [52] indicated that while Africa is statistically different than all the other continents (p-value: <0.001 in all cases), there exist no significant differences between each pairing of the other three continents (Americas vs. Asia p-value: 0.204, Americas vs. Oceania: p-value: 0.568, p-value: Asia vs. Oceania: 0.816). Within each continent, there exist further differences among the subregions. No statistically significant difference was found between each of the three subregions in Africa (Eastern vs. Middle p-value: 0.914, Eastern vs. Western p-value: >0.999, Middle vs. Western p-value: 0.886). However, in Asia, there was a statistically significant difference between the mean COI in South-Eastern Asia and Southern Asia (p-value: 0.0282).

We found a positive correlation between the COI and the microscopy prevalence (Table 3). In regions with a lower prevalence, there were few samples with a COI larger than 2. In fact, in regions with a prevalence less than or equal to 0.01, more than 95% of samples had a COI of 1 or 2. In regions with a higher prevalence, there were more samples with a higher COI. In particular, in regions where the prevalence was greater than or equal to 0.1, more than 20% of samples had a COI greater than 2.

## Discussion

Despite advances in sequencing technologies and the development of various methods for estimating the COI, no direct measures have been developed to rapidly estimate the COI. In this work, we present two novel methods for directly estimating the COI based on minor allele frequencies. Our methods can provide rapid and accurate estimates. Compared to the current state-of-the-art estimation software, *THE REAL McCOIL*, *coiaf* generated similar point estimates faster, even when conducting 100 bootstrap replicates. The ability to produce point estimates in under a second can provide researchers with immediate information on the COI. Moreover, the performance of our methods suggests that they can be scaled to use whole-genome data with

smaller increases in computational time than exhibited by *THE REAL McCOIL*. Our methods also provide a continuous measure that may provide insight into relatedness.

Through several simulations, we further explored how changing key sequencing variables, such as the number of loci and read depth at each locus, altered our software's performance. We showed that for samples with a low and moderate COI, our methods could accurately predict the COI even with low coverage and a small number of loci. However, as the COI increased, these parameters became more important—a lack of sufficient sequencing corresponded with an underprediction of the true COI. Additionally, we demonstrated that several important factors could influence results, such as sequencing error or overdispersion in parasite density. Importantly, we also show that the population mean WSMAF is an unbiased estimator of the PLMAF (see Appendix C.1 in S1 Appendices). This finding requires certain assumptions which may not be met, for example, for loci associated with drug resistance and when sampling selectively from individuals following drug treatment. Nevertheless, this provides further advantages to using allelic read depth for COI estimation rather than haplotype calls, which are known to lead to biased estimates of the population-level allele frequency if the COI of samples is not accounted for [53].

An application of *coiaf* on several thousand *P. falciparum* samples from malaria-endemic countries in four continents from 2002–2015 [42] resulted in a comprehensive map of the complexity of infection worldwide. This study builds on previous reviews of the distribution of the COI globally [54] and is the first study to our knowledge to provide a holistic view of the COI based on allelic read depth as opposed to traditional methods leveraging *msp1* and *msp2* haplotyping.

In general, our results were in agreement with previously reported findings. For instance, we estimated lower average COI values in areas with historically lower malaria prevalence, such as South America and Southern Asia. In particular, in the Americas and Asia, we report a mean COI of 1.03 and 1.26, respectively. Previous efforts to estimate the COI in these regions have also found similarly low values. While directly comparing our estimates against these would be incorrect as the date and location of sample collection are very different, we are encouraged by the similarity in estimates. For example, in Brazil, the COI has been previously measured as low as 1.1 in the early 1990s [55]. In Papua New Guinea [40, 56, 57], Bangladesh [58], and Malaysia [59], previous estimates of the COI range between 1.00 and 2.12, 1.22 and 1.58, and 1.20 and 1.37, respectively. In Africa, on the other hand, we were surprised to find little to no difference in average COI estimates between the three subregions we studied: Eastern Africa, Middle Africa, and Western Africa, despite large differences in average malaria prevalence in these regions. In contrast, previous studies of the COI in African settings have found higher average COI values in some regions and, in general, greater variability. For example, in Cameroon, large mean COIs between 2.33 to 3.82 have been reported [60–62]. Conversely, in Ghana, the mean COI has been reported to be between 1.13 and 1.91 in 2012–2013 [63].

Much of the work surrounding the complexity of infection is motivated by the fact that COI has been proposed as an indicator of transmission intensity. Unfortunately, the relationship between COI and malaria prevalence remains an area of much debate, with many individual studies finding different relationships [64, 65]. Lopez and Koepfli report in their review article that across the 153 studies examined, there was a weak correlation between the mean COI and prevalence, an observation that agrees with our findings [54]. As previously noted, multiple patient-level factors (e.g., age and clinical status) may affect the relationship between malaria prevalence and COI. Additionally, Karl et al. suggested that this weak correlation may be attributed to spatial effects and the existence of geographic "hotspots," where transmission may be much higher than in surrounding areas, causing individuals to have a greater COI [66]. Moreover, multiple studies have highlighted that seasonality affects the observed COI

[67, 68]. Consequently, while there is undoubtedly a relationship between transmission intensity and COI, it is important to be aware of how many factors (age, clinical status, seasonality, spatial effects, parasite density, time since the last infection, and methods of detecting multiple infections) may impact this relationship. For example, metadata is only available while analyzing the data from the MalariaGEN *Plasmodium falciparum* Community Project [42] regarding the year and location of sample collection. Without being able to account for the other factors that impact the COI, we cannot make more meaningful interpretations of our analysis of COI patterns. Lastly, it is worth recalling that malaria prevalence is not directly related to transmission intensity. For example, two regions with the same malaria prevalence will likely have different transmission intensities if intervention coverage differs between the region. Therefore, the variance observed between malaria prevalence and COI may reflect that malaria prevalence is itself an imperfect predictor of transmission intensity.

Our work is not without limitations. In part, limitations stem from our methods relying on certain biological assumptions, which may not be met in the real world. Additionally, the accuracy of our algorithms is impacted by sequence error. While this was not an issue in our analysis of the MalariaGEN *Plasmodium falciparum* Community Project (see Fig AK in S1 Appendices), high levels of sequence error need to be monitored and accounted for. While our software package allows users to account for this by providing a level of suspected sequencing error, sequence error is unlikely to be constant across the genome, and accurate inference of sequencing error is an active research challenge [69]. Furthermore, while our methods account for the coverage at each locus, if there is a low overall coverage for a sample, our results may underpredict the true COI. Similarly, many modern genotyping approaches rely on a pre-amplification step before genotyping or sequencing. The amplification of higher parasitemia samples may obscure lower prevalence parasite strains and result in lower inferred COI estimates. Lastly, our methods assume that the population-level minor allele frequency (PLMAF) is well captured by the samples provided. Sampling bias, undersampling, or complex spatial patterns of allele frequencies resulting from the spatial landscape of malaria transmission and heterogeneities in transmission intensities in a region may result in an inaccurate PLMAF estimation, which may influence our estimated COI.

In conclusion, we developed two direct measures for estimating the COI given the within-sample allele frequency of a sample and the population-level allele frequency. We also add a new understanding of how population-level allele frequencies can be estimated without bias by relying on continuous estimates of within-sample allele frequencies. Our methods could estimate the COI for samples in less than a second and were accurate compared to simulated data and current COI estimation techniques. Our software will aid in estimating the complexity of infection, an increasingly important population genetic metric for inferring malaria transmission intensity and evaluating malaria control interventions [18, 70].

## Supporting information

**S1 Appendices. Appendices.** Includes the complete problem formulation, including derivations of the Variant Method and the Frequency Method, additional information about our methods and software package, and all omitted figures.
(PDF)

## Acknowledgments

We thank the MalariaGEN *Plasmodium falciparum* Community Project for maintaining a large collection of sequencing data and variant calls.

## Author Contributions

**Conceptualization:** Oliver J. Watson, Ozkan Aydemir, Robert Verity, Jeffrey A. Bailey.

**Data curation:** Aris Paschalidis, Oliver J. Watson.

**Formal analysis:** Aris Paschalidis, Oliver J. Watson.

**Funding acquisition:** Robert Verity, Jeffrey A. Bailey.

**Methodology:** Aris Paschalidis, Oliver J. Watson, Ozkan Aydemir, Robert Verity, Jeffrey A. Bailey.

**Software:** Aris Paschalidis, Oliver J. Watson.

**Supervision:** Jeffrey A. Bailey.

**Visualization:** Aris Paschalidis, Oliver J. Watson.

**Writing – original draft:** Aris Paschalidis, Oliver J. Watson.

**Writing – review & editing:** Aris Paschalidis, Oliver J. Watson, Ozkan Aydemir, Robert Verity, Jeffrey A. Bailey.

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
