## [Decision Letter · Decision Letter 0]

25 Jul 2022

Dear Dr Bailey,

Thank you very much for submitting your manuscript "coiaf: directly estimating complexity of infection with allele frequencies" for consideration at PLOS Computational Biology.

As with all papers reviewed by the journal, your manuscript was reviewed by members of the editorial board and by several independent reviewers. In light of the reviews (below this email), we would like to invite the resubmission of a significantly-revised version that takes into account the reviewers' comments.

In particular, the the reviews, as a whole, suggest important directions that would be valuable for the authors to explore, particularly (i) evaluating and discussing when/whether the complexity/memory advantages of coiaf are retained if one chooses to draw bootstrapped samples, (ii) the circumstances under which compute & memory are critical, and (iii) the conditions required for statistical consistency. I also thank the authors for submitting a paper whose clarity of writing allowed all reviewers and editor to engage fully with the manuscript. 

We cannot make any decision about publication until we have seen the revised manuscript and your response to the reviewers' comments. Your revised manuscript is also likely to be sent to reviewers for further evaluation.

Sincerely,

Daniel B Larremore, Ph.D.

Associate Editor

PLOS Computational Biology

Thomas Leitner

Deputy Editor

PLOS Computational Biology

Reviewer's Responses to Questions

**Comments to the Authors:**

Reviewer #1: Uploaded as attachment

Reviewer #2: Uploaded as attachment

Reviewer #3: In this manuscript Pashalidis and colleagues present a new method for estimating the complexity of infection (COI) of Plasmodium infections. COI is a foundational metric in Plasmodium genetics/genomics, and its estimation is an integral part of all bioinformatic analyses. In addition, as the authors discuss, it is increasingly being considered as part of the genomic epidemiology toolkit for tracking changes in transmission. In this clear, well written manuscript, the authors give a good overview of current tools in the field, including their strengths and limitations. They then present their own method, packaged as coiaf, as being a faster and more scalable alternative to the current tool of choice, TheReaMcCoil.

Currently, the comparison between TheRealMcCoil and coiaf is limited to sequence data from a set of globally distributed samples. The two methods perform similarly but not identically, and I am left wondering how the accuracy of the two approaches compares. I suggest that the authors include a head-to-head comparison of coiaf and TheRealMcCoil on the simulated data as well, showing both COI and population level allele frequency estimates for the two methods.

In regards to assessing accuracy, I also suggest that the authors expand the simulated data results presented in the main text. I appreciated the wider range of parameters explored in the Supplement, and readers would benefit from seeing these data in the main results. In particular, I think it is worth highlighting the effects of strain proportions, read depth, and overdispersion of read counts.

Fig 1 in the appendix shows that coiaf produces an unbiased estimator of population level allele frequencies, however, accuracy is low, especially in the region of typical sample sizes (<=100). I would like to see these estimates compared to those produced using current common approaches: (1) estimating allele frequencies from putatively monoclonal samples only and (2) using TheRealMcCoil. Whether highly accurate maf estimates are necessary for COI estimation is another matter. The authors could demonstrate that their method provides sufficient accuracy for estimating COI without the added computational expense of accurately estimating population mafs. In this case, however, I would suggest they clarify this in the text.

In the absence of evidence that coiaf is more accurate than TheRealMcCoil, coiaf's increased computational efficiency strikes me as the most compelling benefit over the other. I would suggest highlighting this in the main Results (with a figure even). The authors might also consider giving increased emphasis to coiaf's ability to handle whole-genome data, although given the diminishing returns observed after crossing the 1000 loci threshold, this may not be a major benefit.

The R package itself was easy to install and run (although a few small comments are below).

In sum, I find this new approach for estimating COI in Plasmodium infections to be compelling and am particularly impressed by the speed and scalability. I, however, suggest a more thorough head-to-head comparison against TheRealMcCoil and a greater exploration of parameters like strain proportions, read depth, and read overdispersion in the main text. This is needed to present a strong case for coiaf becoming a new tool of choice.

Small comments and suggestions:

- The authors state that continuous COI estimates can provide insight into the relatedness of parasites in complex infections. In theory, I agree with this; in practice, I do not know how easy it is to interpret such numbers vis-a-vis other factors like low read coverage and genotyping error. Simulating and analyzing data to this effect would be a useful addition to this manuscript as I agree that this is an area of interest for the field.

- In Fig 1, the points are so dense that it is hard to derive any information from these plots. If the goal is to visualize the analytical steps, a cartoon schematic might be easier to interpret. (This is just a suggestion)

- The DePloidIBD paper ( https://doi.org/10.7554/eLife.40845) does incude information on the global distribution of Ke ('effective number of strains'). While this isn't equal COI in a strict sense, it is highly related. These results could be compared to the global COI estimates made here.

- Judging from the performance of the script, "minor allele freq" is really "non-reference allele freq"

- It appears the simulations assume all loci are unlinked. This is a reasonable and common assumption, but I suggest making this explicit in the text since WGS and some targeted sequencing approaches generate linked variants. (I did see that LD was addressed in the appendix in regards to running TheRealMcCoil.)

- Many genotyping approaches now rely on a pre-amplification step prior to genotyping/sequencing. The (potential) effects of this could be mentioned in the Discussion.

- Do you suggest that users take the COI probabilities into account? Does some filtration at this level increase accuracy?

- On the GitHub page, the links to the vignettes are broken

- A simple manual would be a helpful addition. For instance, the text mentions incorporating genotyping error into the model, but it was unclear from the examples how to do this short of editing the main code. I also never found the code used for bootstrapping CIs.

- Running the 'method 2, continuous' example in 'example_real_data.Rmd', all COI values were over-inflated (including many with COI=25)

I found a few small errors in 'example_real_data.Rmd' (NB: I was using the latest release, not the dev version)

line 57, 85, 100, 128: a character string [0,1] is requested for coi_method

line 56, 84, 99, 127: expected tibble values were wsaf and plaf

**Have the authors made all data and (if applicable) computational code underlying the findings in their manuscript fully available?**

Reviewer #1: Yes

Reviewer #2: Yes

Reviewer #3: Yes

PLOS authors have the option to publish the peer review history of their article (what does this mean?). If published, this will include your full peer review and any attached files.

Reviewer #1: No

Reviewer #2: No

Reviewer #3: No
---

## [Decision Letter · Decision Letter 1]

9 Feb 2023

Dear Dr Bailey,

Thank you very much for submitting your manuscript "coiaf: directly estimating complexity of infection with allele frequencies" for consideration at PLOS Computational Biology. As with all papers reviewed by the journal, your manuscript was reviewed by members of the editorial board and by several independent reviewers. The reviewers appreciated the attention to an important topic. Based on the reviews, we are likely to accept this manuscript for publication, providing that you modify the manuscript according to the review recommendations.

Sincerely,

Daniel B Larremore, Ph.D.

Academic Editor

PLOS Computational Biology

Thomas Leitner

Section Editor

PLOS Computational Biology

Reviewer's Responses to Questions

**Comments to the Authors:**

Reviewer #1: uploaded as an attachment

Reviewer #2: The authors have sufficiently addressed my major concerns through direct comparisons to THE REAL McCOIL on simulated data. I only have minor comments regarding some of the supplementary figures and minor copy editing.

1. line 118 appears to have the beginning cut off

2. in figures comparing performance with THE REAL McCOIL using simulated data, how many bootstrapped samples are being used to estimate 95% CI for coiaf methods? If it's not default setting of 100, then this should be mentioned and the plots comparing speeds should be done with the same number of bootstrapped samples

3. It's unclear to me the difference between epsilon and sequencing error in simulations. I see that epsilon is varied when evaluating the performance of the different estimation approaches and the figure text (L.1 fig 10) explains that sequencing error is fixed at 1%. However, in the comparisons to THE REAL McCOIL (L.1.1 fig 5), the text states that the sequencing error is varied, is this distinct from varying epsilon? I ask because the values chosen varying sequencing error when comparing to THE REAL McCOIL match the values of epsilon varied earlier (0.05, 0.01, 0.015), however the performance is wildly different for coiaf between the two figures. This same concern applies to estimating PLAF as well, but that is not evaluated earlier.

Reviewer #3: In this revision, Paschalidis and colleagues have given thoughtful consideration to reviewers comments and substantially strengthened this version. In my opinion, this current manuscript clearly presents sufficient detail for readers to understand and evaluate this new COI estimation method in relation to currently used approaches.

**Have the authors made all data and (if applicable) computational code underlying the findings in their manuscript fully available?**

Reviewer #1: Yes

Reviewer #2: Yes

Reviewer #3: Yes

PLOS authors have the option to publish the peer review history of their article (what does this mean?). If published, this will include your full peer review and any attached files.

Reviewer #1: No

Reviewer #2: No

Reviewer #3: No

Figure Files:

Data Requirements:

Reproducibility:

References:

---

## [Editor Report · Decision Letter 2]

1 May 2023

Dear Dr Bailey,

We are pleased to inform you that your manuscript 'coiaf: directly estimating complexity of infection with allele frequencies' has been provisionally accepted for publication in PLOS Computational Biology.

Best regards,

Daniel B Larremore, Ph.D.

Academic Editor

PLOS Computational Biology

Thomas Leitner

Section Editor

PLOS Computational Biology

---

## [Editor Report · Acceptance letter]

5 Jun 2023

PCOMPBIOL-D-22-00780R2 

coiaf: directly estimating complexity of infection with allele frequencies

Dear Dr Bailey,

I am pleased to inform you that your manuscript has been formally accepted for publication in PLOS Computational Biology. Your manuscript is now with our production department and you will be notified of the publication date in due course.

With kind regards,

Bernadett Koltai
